# An Evaluation of Factors Influencing the Resilience of Flood-Affected Communities in China

Wenping Xu [1,2], Yingchun Xie [1], Qimeng Yu [1] and David Proverbs [3,*]

1   School of Management, Wuhan University of Science and Technology, Wuhan 430065, China
2   Center for Service and Engineering, Wuhan University of Science and Technology, Wuhan 430065, China
3   Faculty of Science and Engineering, University of Wolverhampton, Wolverhampton WV1 1LY, UK
*   Correspondence: david.proverbs@wlv.ac.uk

**Abstract:** In recent years, the acceleration of urbanization processes coupled with more frequent extreme weather including more severe flood events, have led to an increase in the complexity of managing community flood resilience. This research presents an empirical study to explore the factors influencing community flood resilience in six communities located in the Hubei Province of China. The study presents the development of a flood resilience evaluation index system, comprising the use of the decision-making trial and evaluation laboratory (DEMATEL) and interpretative structural modeling method (ISM) methods. The results show that the three most important factors affecting the flood resilience capacity of the community are (i) the investment in disaster prevention, (ii) disaster relief capacity and (iii) flood control and drainage capacity. The differences between the six communities were analyzed across four dimensions to reveal the strengths and weaknesses of the communities across these dimensions and in terms of their overall resilience. By analyzing the causal hierarchical relationship that affects community flood resilience, this study helps to enhance community resilience to flood disasters and reduce disaster risk. These findings are conducive to enhancing the sustainable development of urban communities and are expected to provide scientific guidance for community risk management and strategic decision-making.

**Keywords:** community flooding; DEMATEL-ISM; TOPSIS





## 1. Introduction

In recent years, the rapid development of urbanization coupled with a changing climate and more frequent intense rainstorms have led to an increase in urban flooding [1]. Flooding is known to have a long-term negative impact on the development of society as a whole and social stability, mainly in terms of economic losses and casualties. In 2020, there has been a number of urban flooding incidents caused by heavy rainfall in China; according to the annual flood and drought report of the People's Republic of China (PRC, 2020), the country experienced its worst flood since 1998, causing great losses. For example, a total of 21 floods occurred in major rivers, leading to a direct economic loss of 0.26% in the GDP ($38.194 billion). The community is the main victim unit in urban flooding events and therefore, community flood resilience research is inextricably linked to urban flooding.

As global warming and human activities intensify, extreme rainfall events will become more frequent in the future, leading to the threat of more deaths and losses [2]. As the most basic unit of a city and the first basic line of defense in disaster response, communities can be highly vulnerable and sensitive to such natural hazards. Strengthening the ability of communities to resist the impacts of flooding and improving awareness and understanding of disaster prevention and response are known to be keys to improved resilience. As such, in recent years, the ability of communities to become more resilient to flooding has become a focus of attention in both practice and in research [3–8].

Currently, there is no uniform definition of resilience in academia and different scholars have different focuses [9]. Resilience initially refers to the ability of a metal to remain stable

or return to its original state when subjected to an external impact. Holling first introduced resilience to the ecological and environmental fields in 1973, considering it as the ability of a system to detect and resolve external shocks in the event of a crisis, while maintaining its primary function [10]. With the in-depth research of scholars, the applicable fields and connotations of resilience have been enriched. The current phase on resilience is dominated by three major areas: engineering resilience [11], ecological resilience [12] and socioecological system resilience [13]. Resilience is increasingly being applied in the field of urban cities and has been the subject of much in-depth research by scholars in various countries. However, more recently, the research focus has gradually shifted from resilient cities to an emphasis on supporting the development of resilient communities. Hence, improving community resilience has become a core component of disaster risk development programs in recent years [14].

There exists a range of resilience concepts proposed by scholars from the perspective of disaster prevention and mitigation. The concept of disaster resilience comprises the various measures and methods that people take to mitigate the damage caused by natural disasters in the process of their occurrence, with the aim of reducing or avoiding the damage caused by natural disasters. For example, in 1999, Mileti defined resilience as the ability of a region to withstand extreme natural events without devastating loss and destruction, while maintaining productivity and normal life, and without the need for substantial out-of-area assistance [15]. Tobin identified resilience as a social organizational structure that minimizes the impact of disasters while being able to quickly restore socioeconomic viability [16]. Godschalk et al. put forward the concept of a resilient city in 2002. They defined a resilient city as a sustainable physical system or human community capable of responding to extreme events, including the ability to survive and function under extreme stress [17]. The U.S. Department of Homeland Security proposes that resilience is an asset, system or network that, in the event of a particular emergency, is capable of performing at set target functional levels and the ability to efficiently mitigate the degree and duration of damage to a system caused by a disaster (or emergency) [18]. The UNISDR defines resilience as the ability of an exposed system, community or society to withstand, absorb, adapt and recover from hazards in a timely and efficient manner, including protecting and restoring its essential elements [19]. In summary, while the definitions of scholars have their own focus, in this research, "resilience" is tentatively defined as the ability to resist risk and regain equilibrium in a short period of time following sudden external damage or threats.

Community resilience is the ability of a community to cope with and recover from an event such as a flood without relying entirely on external support so that the community can quickly return to a healthy state. Community resilience emphasizes the ability of communities to become more independent in coping with and recovering from disasters [20]. When conducting research on resilient communities, community resilience is often analyzed by constructing a framework. For example, De Iuliis et al. applied the PEOPLES framework to hierarchize the impact indicators of community resilience when measuring and improving community resilience [21]. Bruneau et al. proposed the 4R elastic resilience framework, a framework that defines quantitative measures of community resilience and contributes to research efforts to improve resilience [22]. Zhang et al. proposed an improved conceptual framework for the analysis of community resilience that integrates the principles of building socioecological resilience and provides a step-by-step process for analyzing community resilience [23]. Each of these frameworks has its own merits, but they are not sufficiently applicable to community flood resilience. Therefore, this paper identifies and validates the factors influencing community flood resilience by constructing a framework and applying the DEAMTEL-ISM and TOPSIS methods.

Moreover, scholars have developed a number of tools to facilitate conducting community resilience assessments. For example, Pfefferbaum et al. developed the Community Advancing Resilience Toolkit (CART), a driving tool for studying community resilience [24]. Tan et al. used an indicator tool developed by FEMA to guide resilience-building initiatives and for managers to use to help build resilience in their communities [25]. These tools

contain many different types of indicators, such as those under the change potential dimension, and therefore are not applicable in this paper; there is a need to select the appropriate tool for the specific situation in practical application.

In order to enhance community flood resilience, scholars have conducted extensive research on flood resilience evaluation. Many studies calculate resilience by constructing an index evaluation system and then weighing the indicators. Moghadas et al. constructed an evaluation index system from six dimensions: social, economic, institutional, infrastructure, community capital and environment. The analytic hierarchy process (AHP) and Technique for Order Preference by Similarity to an Ideal Solution (TOPSIS) models were used to prioritize and evaluate resilience in Tehran [26]. Based on the three attributes of resistance, resilience and adaptability, Chen et al. established an evaluation index system for urban resilience in the context of storm water disasters, constructed a KL-TOPSIS comprehensive evaluation calculation model and evaluated the urban resilience of the city of Wuhan during different periods [27]. Li Ya et al. constructed an evaluation index system of urban disaster resilience for the six aspects of economy, society, environment, community, infrastructure and organization and evaluated the resilience level of 288 prefecture-level cities in China. Through the reference literature and research findings, the four aspects of community capital, infrastructure, economic development and good disaster prevention and mitigation were selected to construct an index system that combines resilience evaluation and disaster prevention and mitigation to influence the judgment of key factors of community resilience to flooding.

Some scholars have also studied the occurrence process and action path of flood disasters [28]. For example, Chen et al. summarized the research results of the characteristics, processes and mechanisms of social vulnerability by studying the relevant literature on social vulnerability and resilience of community flood disasters [1]. Wu et al. evaluated and graded the flood disaster resilience of 76 cities in the middle and lower reaches of the Yangtze River in four stages: resistance, early warning, response and recovery [29]. Unlike these scholars, this paper studies the usual state of community resilience to floods, not during the process of flood disasters, but during the resilience phase of community flood resilience. By identifying the key factors affecting community resilience to floods, this paper gives managers certain suggestions to prompt them to make adjustments to the community to improve community resilience to floods.

Many methods can be used to sort and prioritize indicators, including the AHP-fuzzy number method [30], the DEMATEL method [31] and the ISM method [32]. Different methods have their own advantages; for example, the AHP-fuzzy number method can compare the index priority, but mainly relies on the use of experts to score indicators based on their experience and expertise and therefore, the results are to some extent subjective. The DEMATEL method can analyze the causal relationship and importance of indicators, but for interrelated multiple indicators, the internal action path cannot be clearly explained. In conclusion, all of these methods have shortcomings when used alone, so we chose to use a combination of the DEMATEL-ISM method, as well as the TOPSIS method. The key factors affecting community resilience to flooding were first identified and then validated by conducting an example analysis. Before using these methods, it was necessary to first establish a community structure and hierarchy model to resist urban floods. This study proposes a comprehensive evaluation model to enhance community resilience to urban floods, covering the following objectives:

(i)　Determine an index of community resilience against urban flooding;
(ii)　Establish a hierarchical structural model to analyze the internal interaction of indicators;
(iii)　Determine the ability of each community to resist flooding and analyze the scores of different communities.

The rest of this article is structured as follows. Section 2 introduces the new framework and quantitative methods, and then in Section 3, the index system and related data analysis

are described. Section 4 presents the results of six case studies of communities in China. The discussion and conclusions are presented in Sections 5 and 6, respectively.

## 2. Research Methods

The structural framework of this study is shown in Figure 1. Firstly, through expert surveys and a review of the literature, the key indicators affecting community flood resilience were identified. The two methods of the decision-making laboratory method (DEMATEL) and interpretative structural model (ISM) were used to analyze the interaction between indicators and draw the network structure model. Finally, the TOPSIS method was used to calculate and rank the resilience of the six communities.

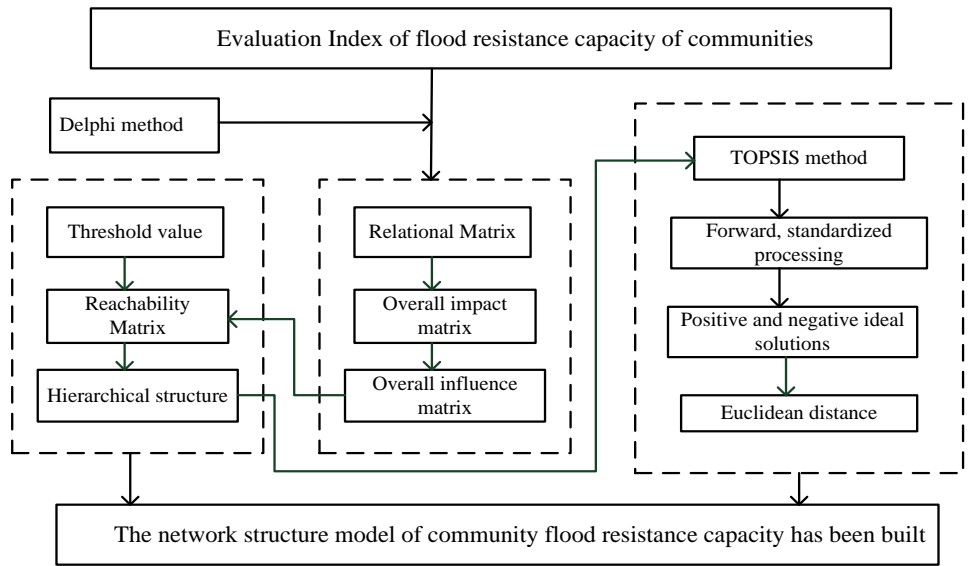

**Figure 1.** Network structure model of the flood resistance capacity of communities.

### 2.1. DEMATEL Method

The decision-making trial and evaluation laboratory (DEMATEL) method focuses on the use of matrix theory and graph theory to assess the strength of the influence of key factors. These factors are analyzed by measuring the center degree, cause degree and influence degree of each influencing factor [31,33]. As the relationship between these internal factors is complex, the DEMATEL method is more suitable for quantitative verification. The DEMATEL method was implemented as follows:

Step 1: Using a matrix to quantify the relationship between the elements to build the initial impact matrix: $A(A = (a_{ij})_{n \times n})$.

$$A = \begin{pmatrix} a_{11} & \cdots & a_{1n} \\ \vdots & \ddots & \vdots \\ a_{n1} & \cdots & a_{nn} \end{pmatrix} \tag{1}$$

Factor $a_{ij}$, $(i = 1, \ldots, n, j = 1, \ldots, n, i \neq j)$, in matrix $A$, represents the degree of direct influence of factor $a_i$ on $a_j$, if $i = j$, $a_{ij} = 0$.

Step 2: According to the original relation matrix $A$, the normalized direct impact matrix $B(B = (b_{ij})_{n \times n})$. The calculation formula is as follows:

$$B = \frac{A}{\max \sum_{j=1}^{n} a_{ij}} \tag{2}$$

In the above equation, it is known that $0 \leq b_{ij} \leq 1$ and $\max \sum_{j=1}^{n} b_{ij} = 1$.

Step 3: The comprehensive influence matrix is denoted as *C*.

$$C = B + B^2 + \cdots + B^n = B(I - B)^{-1} = (C_{ij})_{n \times n} \tag{3}$$

where I is the unit matrix and $C_{ij}$ is the element of the synthetic influence matrix.

Step 4: The elements of matrix *C* are summed horizontally to obtain the degree of influence $D_i$ of the corresponding element, and the elements of matrix *C* are summed column-wise to obtain the degree of being influenced $E_i$ of the corresponding element [34]. According to the comprehensive influence matrix *C*, the influence degree $D_i$, the affected degree $E_i$, the center degree $D_i + E_i$ and the reason degree $D_i - E_i$ of each factor are obtained.

$$D_i = \sum_{j=1}^{n} c_{ij}, i = 1, 2, 3, \cdots, n \tag{4}$$

$$E_i = \sum_{j=1}^{n} c_{ji}, i = 1, 2, 3, \cdots, n \tag{5}$$

*2.2. ISM Method*

ISM is a systematic scientific method with wide application and originates from structural modeling. ISM first sorts the systems to be analyzed, then decomposes them into seed systems, then analyzes each factor and also the relationship between each factor. On this basis, the conceptual model is transformed into a directed graph, and the basic structure of the system is finally revealed by Boolean logic operation and expressed in the form of the simplest hierarchical directed topological graph without affecting the function of the whole system [32,35]. Compared with tables, words and mathematical formulas, ISM has a huge advantage in describing the nature of a system. The result is a hierarchical topology, which is more concise.

(1)    Calculate the overall impact matrix.

The overall impact matrix reflects the overall interaction between the various indicators, including the impact emanating from itself. Thus, the overall impact matrix *D* can be calculated as follows:

$$D = C + I = (d_{ij})_{n \times n} \tag{6}$$

where I is the unit matrix.

(2)    Establish the reachable matrix.

Set the threshold $\lambda$ and then, the reachable matrix is determined by formula (7):

$$d_{ij} = \begin{cases} 1, d_{ij} \geq \lambda, i = 1, \dots, n \\ 0, d_{ij} < \lambda, i = 1, \dots, n \end{cases} \tag{7}$$

where "1" represents a strong correlation between the two indicators, and "0" represents a weak or rare correlation.

(3)    Determine the hierarchical evaluation network.

The antecedent set $A_i$ and reachable set $R_i$ can be determined by Equations (8) and (9):

$$A_i = \{a_i \mid a_i \in A, k_{ij} \neq 0, i = 1, 2, \cdots, n\} \tag{8}$$

$$R_i = \{a_i \mid a_i \in A, k_{ij} \neq 0, i = 1, 2, \cdots, n\} \tag{9}$$

If "$R_i \cap A_i = R_i$" is satisfied, the reachability set of $R_i$ is completely included in the antecedent set of $A_i$. In other words, for all indicators in $R_i$, the antecedents of $S_i$ can be found in $A_i$. Other indicators can reach the indicator $S_i$, while $S_i$ cannot reach other factors. Accordingly, all elements in $R_i$ form the set of indicators, and the corresponding row and column are deleted from the matrix *D*. Class division is used to divide different indicators into different levels. Finally, the hierarchical structure network can be determined.

### 2.3. TOPSIS Method

The Technique for Order Preference by Similarity to an Ideal Solution (TOPSIS) method was first developed by Hwang and Yoon in 1981 [36]. It is a method of ranking a finite number of evaluation objects according to their closeness to an idealized goal. The TOPSIS method can also be simply called a ranking method for identifying the optimum solution, and the results can accurately reflect the differences among the evaluation programs. The steps are as follows:

(1) Forward process the reverse index:

$$X_{ij} = \max - X_{ij} (1 \le i \le m, 1 \le j \le n) \tag{10}$$

(2) Use the vector gauge method to obtain gauge matrix $G$:

$$G = \left[g_{ij}\right] = \left[x_{ij} / \sqrt{\sum_{i=1}^{m} x_{ij}^2}\right] (1 \le i \le m, 1 \le j \le n) \tag{11}$$

(3) Determine the positive ideal solution and the negative ideal solution.

A positive ideal solution is a vector consisting of the maximum values in each column of the $Z$ matrix:

$$Z^+ = \left[z_1^+, z_2^+, \cdots, z_n^+\right] = \left[\max(z_{11}, z_{21}, \cdots, z_{m1}), \max(z_{12}, z_{22}, \cdots, z_{m2}), \cdots, \max(z_{1n}, z_{2n}, \cdots, z_{mn})\right] \tag{12}$$

A negative ideal solution is a vector consisting of the smallest values in each column of the $Z$ matrix:

$$Z^- = \left[z_1^-, z_2^-, \cdots, z_n^-\right] = \left[\min(z_{12}, z_{22}, \cdots, z_{m2}), \cdots, \min(z_{1n}, z_{2n}, \cdots, z_{mn})\right] \tag{13}$$

(4) Calculate the Euclidean distance between each indicator and the ideal solution:

$$d_i^+ = \sqrt{\sum_{j=1}^{n} \left(z_{ij} - z_j^+\right)^2} (1 \le i \le m, 1 \le j \le n) \tag{14}$$

$$d_i^- = \sqrt{\sum_{j=1}^{n} \left(z_{ij} - z_j^-\right)^2} (1 \le i \le m, 1 \le j \le n) \tag{15}$$

(5) Calculate the relative proximity of each community to a positive ideal solution, which represents the highest flood resilience:

$$S_i \frac{d_i^-}{d_i^+ + d_i^-}, 0 \le S_i \le 1 \ (1 \le i \le m) \tag{16}$$

## 3. Construction of an Evaluation Index System

### 3.1. Index Selection

Based on the literature, four levels were selected from the various indexes of community flood resilience, namely: (i) community capital; (ii) economic development; (iii) infrastructure; and (iv) disaster prevention and mitigation [27,30,31,33,37,38]. A total of 15 indicators was selected across the four levels, as shown in Table 1.

Community capital is a concept proposed by the American scholars, the Floras, which refers to the resources a community possesses, and they are transformed into capital when they are used to generate new resources for investment. The community capital level in this paper includes four indicators: community relations, sense of belonging, community solidarity and organizational activities [39]. When community capital reaches a certain level, the community is highly harmonious and is able to exert a high degree of cohesiveness among its residents when faced with a flood disaster.

Economic development refers to a series of economic developments related to residents in the community such as economic income, which is used to measure the development of

the community. In this paper, the economic development level includes three indicators: income level, health insurance coverage and financial investment in disaster prevention [40]. Better economic development affects the community's basic public infrastructure and improves the community's ability to withstand flooding.

Infrastructure refers to the basic public facilities in the community that help residents exercise or get around as well as take refuge in case of emergency. The infrastructure level includes four indicators: the ability to provide medical services, the ability to provide easy access to transportation, the installation of public shelters and the age of housing [41]. Infrastructure is an important safeguard in the face of an incoming disaster, especially in the face of a sudden situation such as a flood, where infrastructure will affect disaster relief capabilities.

Disaster prevention and mitigation refers to the community's preventive measures for disasters, as well as the community's response to disasters and the reduction in harm to residents when disasters strike. The level of disaster prevention and mitigation includes four indicators: monitoring and early warning capability; disaster relief capability; disaster prevention and education; and flood prevention and drainage capability [42]. Disaster prevention and mitigation is prepared to face disasters in normal times and to hold a vigilant mentality so that when faced with a flood, rescue can be carried out as quickly as possible, as well as to reduce casualties and losses.

**Table 1.** System of influencing factors of community flood resilience.

| Target Layer | Standard Layer | Index | Source | Serial Number |
|---|---|---|---|---|
| System of influencing factors of flood resilience in community | Community capital | Community relations | Chen et al. [43] | $a_1$ |
| | | Sense of belonging | Chen et al. [43] | $a_2$ |
| | | Community mutual assistance | Zhong et al. [44] | $a_3$ |
| | | Organizational activities | Xu et al. [45] | $a_4$ |
| | Economic development | Annual income level | Wang et al. [46] | $a_5$ |
| | | Medical insurance coverage | Chen et al. [27] | $a_6$ |
| | | Disaster prevention fund investment | Chen et al. [33] | $a_7$ |
| | Infrastructure | Medical service capacity | Xiang et al. [47] | $a_8$ |
| | | Convenient traffic capacity | Chen et al. [43] | $a_9$ |
| | | Public sanctuary setting | Chen et al. [33] | $a_{10}$ |
| | | Housing age | Chen et al. [43] | $a_{11}$ |
| | Disaster prevention and mitigation capability | Monitoring and early warning capability | Zhong et al. [44] | $a_{12}$ |
| | | Disaster rescue capability | Chen et al. [33] | $a_{13}$ |
| | | Disaster prevention publicity and education | Zhong et al. [44] | $a_{14}$ |
| | | Flood control and drainage capacity | Xu et al. [45] | $a_{15}$ |

*3.2. Results of the DEMATEL Method*

(1)    Screening of experts and distribution of questionnaires

Through consultation and investigation with experts who were screened, including representatives from relevant government departments and university researchers in this research direction, their opinions on the relative importance of community resilience indicators were collected through an online questionnaire survey. The survey has strict requirements on the experience of the respondents, requiring experts or scholars to be familiar with community flood resilience research or have relevant work experience. A total of 50 questionnaires was distributed and 36 valid questionnaires were recovered. The 36 respondents were classified according to the four characteristics of their highest education, work unit, work qualification and research understanding level (Table 2).

(2) Expert scoring resulting in an initial matrix *A*

**Table 2.** Characteristics of Respondents.

| Characteristics | Option | Number | Proportion |
|---|---|---|---|
| highest academic credentials | College degree or below | 2 | 6% |
| | Bachelor's degree | 8 | 22% |
| | Master's degree | 20 | 55% |
| | Ph.D. or above | 6 | 17% |
| work seniority | 5 years or below | 6 | 17% |
| | 5–10 years | 16 | 44% |
| | 10–20 years | 10 | 28% |
| | more than 20 years | 4 | 11% |
| work unit | government department | 12 | 33% |
| | higher educational institutions | 11 | 31% |
| | research institution | 8 | 22% |
| | another unit | 5 | 14% |
| Level of research understanding | Not very understanding | 0 | 0% |
| | General understanding | 3 | 8% |
| | Comparative understanding | 16 | 45% |
| | Very understanding | 17 | 47% |

Experts and scholars were invited to score the interaction between the impact indicators as shown in Figure 1 with corresponding scores: 0 is no impact; 1 is impact, but impact is weak; 2 is general impact; 3 is great impact; and 4 is strong impact. The initial relationship matrix *A* was scored and averaged, as shown in Table 3.

(3) Calculation of the initial matrix *A* according to the formula in the DEMATEL method

**Table 3.** Relationship Matrix *A*.

| | $a_1$ | $a_2$ | $a_3$ | $a_4$ | $a_5$ | $a_6$ | $a_7$ | $a_8$ | $a_9$ | $a_{10}$ | $a_{11}$ | $a_{12}$ | $a_{13}$ | $a_{14}$ | $a_{15}$ |
|---|---|---|---|---|---|---|---|---|---|---|---|---|---|---|---|
| $a_1$ | 0 | 3 | 2 | 3 | 2 | 2 | 2 | 2 | 1 | 1 | 2 | 3 | 2 | 3 | 2 |
| $a_2$ | 3 | 0 | 2 | 2 | 2 | 1 | 2 | 2 | 2 | 1 | 2 | 3 | 1 | 2 | 1 |
| $a_3$ | 3 | 2 | 0 | 2 | 1 | 2 | 2 | 2 | 1 | 2 | 2 | 2 | 3 | 3 | 1 |
| $a_4$ | 2 | 3 | 2 | 0 | 2 | 2 | 2 | 1 | 2 | 2 | 1 | 1 | 2 | 2 | 2 |
| $a_5$ | 2 | 2 | 3 | 3 | 0 | 4 | 3 | 2 | 2 | 1 | 2 | 2 | 1 | 3 | 1 |
| $a_6$ | 2 | 1 | 2 | 2 | 2 | 0 | 2 | 3 | 2 | 2 | 2 | 2 | 2 | 1 | 2 |
| $a_7$ | 1 | 2 | 1 | 3 | 2 | 2 | 0 | 3 | 2 | 2 | 3 | 3 | 3 | 3 | 4 |
| $a_8$ | 2 | 3 | 2 | 2 | 2 | 2 | 1 | 0 | 2 | 2 | 2 | 2 | 2 | 2 | 2 |
| $a_9$ | 2 | 1 | 2 | 2 | 1 | 2 | 3 | 3 | 0 | 3 | 3 | 2 | 3 | 2 | 3 |
| $a_{10}$ | 2 | 2 | 1 | 1 | 2 | 2 | 1 | 2 | 1 | 0 | 3 | 2 | 3 | 2 | 2 |
| $a_{11}$ | 2 | 2 | 2 | 2 | 1 | 2 | 2 | 1 | 2 | 2 | 0 | 2 | 3 | 2 | 2 |
| $a_{12}$ | 2 | 2 | 2 | 2 | 1 | 2 | 2 | 2 | 1 | 2 | 2 | 0 | 3 | 2 | 3 |
| $a_{13}$ | 1 | 2 | 2 | 2 | 2 | 2 | 3 | 2 | 2 | 2 | 2 | 2 | 0 | 3 | 4 |
| $a_{14}$ | 2 | 2 | 2 | 2 | 1 | 2 | 2 | 2 | 2 | 1 | 2 | 4 | 3 | 0 | 3 |
| $a_{15}$ | 1 | 2 | 2 | 2 | 2 | 1 | 2 | 1 | 2 | 2 | 1 | 2 | 3 | 3 | 0 |

According to the relation matrix *A*, Formula (2) was used to normalize it and the direct influence matrix *B* was obtained. According to matrix *B*, the comprehensive influence matrix *C* was obtained by using Formula (3), as shown in Table A1. On the basis of the preceding results, using Formulas (4) and (5), the influence degree, the degree of being affected, the degree of centrality and the degree of cause in Table 4 were obtained.

(4) Drawing a diagram from the calculations

**Table 4.** Cause–result elements.

| Index | Influence Degree | Affected Degree | Center Degree | Cause Degree | |
|---|---|---|---|---|---|
| **Community relations ($a_1$)** | **5.558** | **5.000** | **10.557** | **0.558** | Cause element |
| Sense of belonging ($a_2$) | 4.902 | 5.426 | 10.328 | −0.524 | Result element |
| Community mutual assistance ($a_3$) | 5.238 | 5.043 | 10.281 | 0.195 | Cause element |
| Organizational activities ($a_4$) | 4.882 | 5.565 | 10.447 | −0.683 | Result element |
| Annual income level ($a_5$) | 5.773 | 4.341 | 10.114 | 1.432 | Cause element |
| Medical insurance coverage ($a_6$) | 5.056 | 5.153 | 10.209 | −0.097 | Result element |
| Disaster prevention fund investment ($a_7$) | 6.250 | 5.407 | 11.657 | 0.843 | Cause element |
| Medical service capacity ($a_8$) | 5.193 | 5.179 | 10.372 | 0.014 | Cause element |
| Convenient traffic capacity ($a_9$) | 5.935 | 4.544 | 10.479 | 1.390 | Cause element |
| Public sanctuary setting ($a_{10}$) | 4.856 | 4.693 | 9.549 | 0.163 | Cause element |
| Housing age ($a_{11}$) | 5.058 | 5.343 | 10.401 | −0.284 | Result element |
| Monitoring and early warning capability ($a_{12}$) | 5.202 | 5.967 | 11.169 | −0.765 | Result element |
| Disaster rescue capability ($a_{13}$) | 5.761 | 6.325 | 12.086 | −0.564 | Result element |
| Disaster prevention publicity and education ($a_{14}$) | 5.576 | 6.120 | 11.696 | −0.543 | Result element |
| Flood control and drainage capacity ($a_{15}$) | 4.910 | 6.044 | 10.954 | −1.134 | Result element |

It can be seen from Table 4 that the DEMATEL method divides the 15 elements into two elements: cause elements and result elements. The sum of the influence degree and the affected degree of each element is called the centrality of the element, which represents the position and role of the element in the system. The difference between the influence degree and the affected degree is called the cause degree of the element. A cause degree > 0 indicates that the element has a great influence on other factors, which is called the cause factor. A cause degree < 0 indicates that the element is greatly affected by other elements, called the result element. As can be seen in Table 4, there are seven cause factors, namely: Community relations ($a_1$); community mutual assistance ($a_3$); income level ($a_5$); disaster prevention fund investment ($a_7$); medical service capacity ($a_8$); housing age ($a_9$); and convenient transportation capacity ($a_{10}$). The other eight factors are the result factors. The reason-centrality diagram was drawn according to Table 4, as shown in Figure 2.

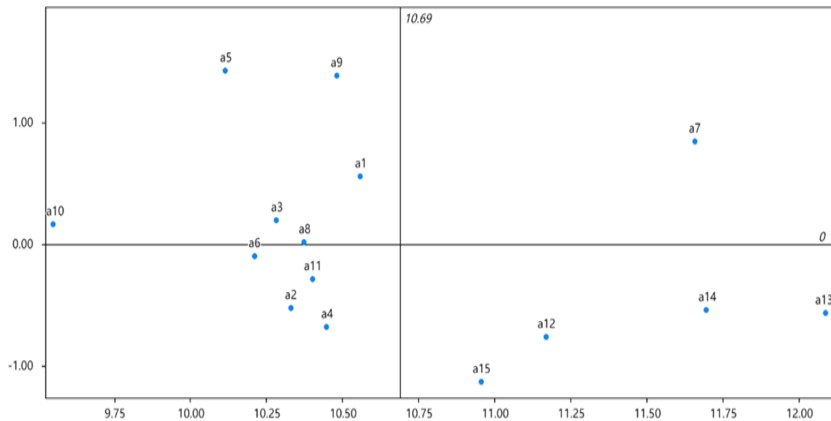

**Figure 2.** Cause–result degree diagram.

According to the analysis described above, a cause–result diagram with the center degree as the abscissa and the cause degree as the ordinate was constructed, as shown in Figure 2. The cause–result diagram can clearly show the cause degree and fruit degree of each factor. The degree of the cause of a factor refers to the influence of this factor on other factors. When the degree of cause is greater than 0, it is proved that this factor is the causal factor and is more able to influence other factors. When the degree of cause is less than 0, this factor is the result factor and as such is more influenced by other factors. The centrality reflects the importance of a factor and the higher the value, the more important it is.

(5)    Analysis of centrality, causality, influence and affectedness

The DEMATEL method allows for the identification of key factors, which can be analyzed from the diagram according to the degree of centrality, the degree of cause, the degree of influence and the degree of being affected, as follows.

Influence degree: the top factors of numerical value are disaster prevention fund investment ($a_7$) and disaster rescue capability ($a_{13}$). These two factors can significantly affect other factors. Through the influence of other factors, these factors affect the flood resilience of the community and are closely related to the overall resilience of the community.

Affected degree: the top factors are disaster rescue capacity ($a_{13}$), disaster prevention education ($a_{14}$) and flood control and drainage capacity ($a_{15}$). These factors are more likely to be affected by other factors. These three elements are very important reference factors for the flood prevention capacity of the community.

Cause degree: income level ($a_5$) has the highest cause, which shows that income level has a strong influence on other factors. However, the impact on each factor is also different; for example, while the income level factor has the greatest impact on medical insurance coverage, it has little impact on community relations and so on.

Centrality: the largest value of the center is the disaster rescue capacity factor ($a_{13}$), which shows that it will have a comprehensive effect on the flood resistance capacity of the community, so attention should be focused on how to improve the disaster rescue capacity of the community. At the same time, it is also the key to improving the disaster response capacity. In short, the disaster rescue capacity of the community should be given full consideration by the government, society and the masses in order to achieve the overall action goal. This will strengthen other factors and help improve the disaster relief capacity.

*3.3. Result of Using the ISM Method*

(1)    Calculate according to the formula and draw a graph based on the results.

Using the comprehensive influence matrix *C* obtained by the DEMATEL method, the overall influence matrix *D* was calculated, the universal threshold λ was used and the reachable matrix *N* was obtained, as shown in Table 5.

Hierarchical division, based on the matrix of reachable matrices, the reachable set R (ai), the linear set A (ai) and the intersection S (ai) were obtained (as shown in Table A2). The first layer was divided according to the previous description, then, the first layer of factors was removed and the remaining factors were stratified for a second time. Based on this process, the 15 factors were eventually divided into five tiers. The final stratification results are shown in Table A2.

Using the ISM method, Figure 3 was developed from the results in Tables 6 and A2, succinctly and intuitively reflecting the interaction among the factors affecting the flood resistance capacity of the built community and defining the hierarchical structure.

(2)    Analyze the stratification of the directed graphs and factors:

    (i)    It can be seen from the ISM model that the 15 factors of the model can be divided into three layers. The sense of belonging, the age of the house and the setting of the public refuge place are located in the first layer of the model, which are the direct influencing factors of the surface layer, indicating that these three factors interact with other factors at this level and have a direct promoting effect on the enhancement of the flood resistance ability of the built community;

    (ii)    The second, third and fourth layers are intermediate influencing factors, including community relations, organizational activities, medical insurance coverage, medical service ability, income level, monitoring and early warning ability, community mutual assistance, convenient transportation ability, disaster prevention publicity, and education. These factors are the means by which the underlying factors work and play a role in linking the factors before and after. Hence, it becomes crucial to strengthen the management of these intermediate factors;

    (iii)    The investment of disaster prevention funds, disaster relief ability, flood control and drainage ability are in the fifth layer, which are the deep fundamental influencing factors and restrict the ability of the built communities to resist flooding.

**Table 5.** Reachability matrix N.

| | $a_1$ | $a_2$ | $a_3$ | $a_4$ | $a_5$ | $a_6$ | $a_7$ | $a_8$ | $a_9$ | $a_{10}$ | $a_{11}$ | $a_{12}$ | $a_{13}$ | $a_{14}$ | $a_{15}$ |
|---|---|---|---|---|---|---|---|---|---|---|---|---|---|---|---|
| $a_1$ | 1 | 1 | 1 | 1 | 0 | 0 | 0 | 0 | 0 | 0 | 0 | 1 | 0 | 1 | 1 |
| $a_2$ | 0 | 1 | 1 | 1 | 0 | 0 | 0 | 0 | 1 | 0 | 0 | 1 | 0 | 1 | 1 |
| $a_3$ | 1 | 1 | 1 | 1 | 1 | 0 | 0 | 0 | 0 | 0 | 0 | 1 | 0 | 1 | 0 |
| $a_4$ | 1 | 1 | 1 | 1 | 0 | 0 | 0 | 0 | 1 | 0 | 0 | 0 | 0 | 1 | 0 |
| $a_5$ | 0 | 0 | 0 | 0 | 1 | 1 | 0 | 1 | 0 | 0 | 0 | 1 | 0 | 0 | 0 |
| $a_6$ | 0 | 0 | 0 | 0 | 1 | 1 | 0 | 0 | 1 | 0 | 0 | 0 | 0 | 0 | 0 |
| $a_7$ | 1 | 1 | 1 | 1 | 0 | 1 | 1 | 1 | 0 | 0 | 1 | 1 | 1 | 1 | 1 |
| $a_8$ | 0 | 1 | 0 | 0 | 0 | 0 | 0 | 1 | 0 | 0 | 0 | 1 | 1 | 1 | 1 |
| $a_9$ | 0 | 1 | 0 | 0 | 0 | 0 | 0 | 0 | 1 | 0 | 0 | 0 | 0 | 0 | 0 |
| $a_{10}$ | 0 | 0 | 0 | 0 | 1 | 0 | 0 | 0 | 0 | 1 | 0 | 1 | 1 | 1 | 1 |
| $a_{11}$ | 0 | 0 | 0 | 0 | 0 | 0 | 0 | 0 | 0 | 0 | 1 | 1 | 1 | 1 | 1 |
| $a_{12}$ | 1 | 1 | 1 | 1 | 0 | 1 | 1 | 1 | 1 | 1 | 1 | 1 | 1 | 1 | 1 |
| $a_{13}$ | 1 | 0 | 1 | 1 | 1 | 1 | 1 | 1 | 1 | 1 | 1 | 1 | 1 | 1 | 1 |
| $a_{14}$ | 1 | 1 | 0 | 1 | 1 | 1 | 1 | 1 | 1 | 1 | 1 | 1 | 1 | 1 | 1 |
| $a_{15}$ | 1 | 1 | 1 | 1 | 1 | 1 | 1 | 1 | 1 | 1 | 1 | 1 | 1 | 1 | 1 |

**Table 6.** Hierarchical decomposition diagram.

| Hierarchy | Elements |
|---|---|
| Layer 1 | $a_2$, $a_9$, $a_{11}$ |
| Layer 2 | $a_1$, $a_4$, $a_6$, $a_8$ |
| Layer 3 | $a_5$, $a_{12}$ |
| Layer 4 | $a_3$, $a_{10}$, $a_{14}$ |
| Layer 5 | $a_7$, $a_{13}$, $a_{15}$ |

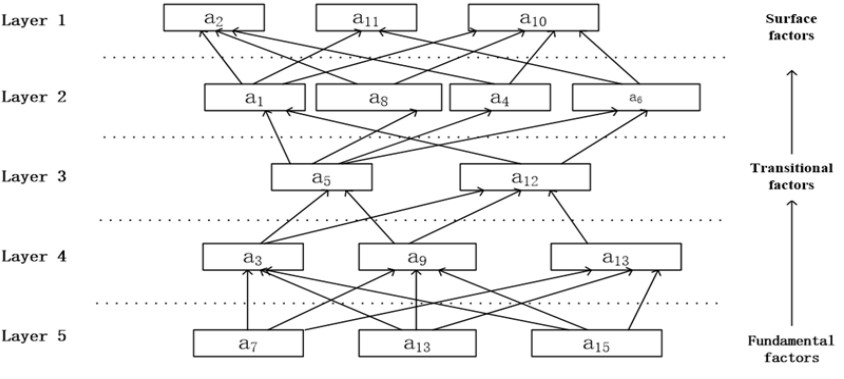

**Figure 3.** Directed graph.

## 4. Empirical Research

### 4.1. Description of Research Area

Hubei Province is located in central China (Figure 4a), surrounded by mountains on the east, west and north, low and flat in the middle, and an incomplete basin slightly open to the south. Hubei is known as the "province of thousands of lakes". There are 755 lakes with a total area of 2983.5 square kilometers. The average annual precipitation of the province in 2021 was 1212 mm. The unique geographical location and climatic characteristics lead to the frequent occurrence of flood disasters [38,48]. For example, in 2016, Hubei Province was hit by heavy rains, and 26 counties and cities reached the level of heavy rain and above, opening the "go out to see the sea" mode, with hourly rainfall of up to 40–60 mm. The areas studied in this research are the communities of Zhangjiawan, Chuanchai, Kangning, Caiyuan, Guanliu and Pearl Garden, which are located in different cities in Hubei Province. The specific locations of the communities are shown in Figure 4b.The characteristics of the six communities are shown in Table A2.

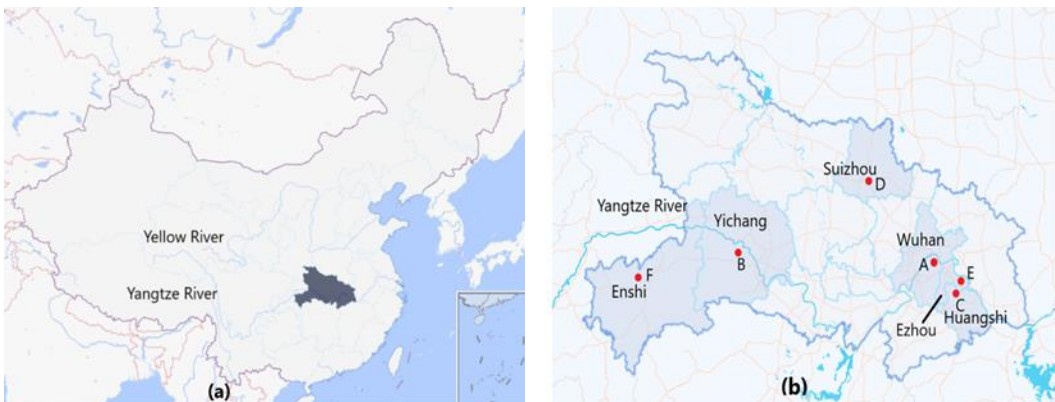

**Figure 4.** (**a**) Location of Hubei Province and (**b**) community location distribution.

### 4.2. Comprehensive Evaluation

The TOPSIS method was used to evaluate the six communities using data from the "Hubei Statistical Yearbook", the National Bureau of Statistics and the Chinese government's network. There are two ways to measure the indicators: one is directly based on specific numbers and specific values can be derived from consulting information, such as income level ($a_5$), financial investment in disaster prevention ($a_7$), housing age ($a_{11}$), etc. The other is to score the indicators that are more abstract and cannot be reflected by specific numbers and assign values to different situations to represent the different performance and impact of each indicator. For example, the concept of community mutual aid is relatively abstract, so the range of 1–5 is used to represent the degree of community mutual aid; factors concerning community relations are difficult to derive from specific values, through the survey found that if there was no conflict between community neighbors, this community score is 1, otherwise it is 0. The specific calculation method and scoring criteria are shown in Table 7 [49].

For example, in Table 7, the $a_1$ factor represents the relationship between neighbors in the community; if there is no conflict between neighbors in the community, the score of the community is 1 and otherwise, it is 0. Similarly, the $a_2$ factor is the willingness to renew the lease or buy a house in the community; if the people who rent or buy a house in the community have the intention to renew the lease or buy a house in the community, the score of the community is 1 and otherwise it is 0. The measure of the $a_3$ factor is the mutual help between neighbors in the community: if there is mutual help between neighbors in the community almost every day, it is 5; if there is mutual help between neighbors, but not every day, though it happens regularly, it is 4; if there is mutual help between neighbors and it happens sometimes, but not regularly, it is 3; if there is mutual help between neighbors, but it happens only once in a while, it is 2; and if there is no mutual help between neighbors

at all, it is 0. The measure of $a_4$ is participation in related organizational activities, and concerns whether the community is a unit that often organizes activities; those who do not participate in any community activity are 0 and the rest are 1.

**Table 7.** Assigning methods.

| Assignment Index | Criteria for Measuring Variables | Assignment |
|---|---|---|
| $a_1$ | Friendly relations among neighbors in the community | Yes = 1; No = 0 |
| $a_2$ | Willingness to renew the lease or buy a house in this community | Yes = 1; No = 0 |
| $a_4$ | Participation in relevant organizational activities | Yes = 1; No = 0 |
| $a_3$ | Neighbors helping neighbors | Always = 5; Often = 4; Sometimes = 3; Occasional = 2; Never = 1 |
| $a_8$ | Time it takes to get to the nearest medical facility | Within 15 min = 4; 15–30 min = 3; 30–60 min = 2; Over 1 h = 1 |
| $a_9$ | Time it takes to get to the nearest bus or subway station | Within 5 min = 3; 5–10 min = 2; 10–15 min = 1 |
| $a_{12}$ | Accuracy of monitoring ability | Always accurate = 5; Often accurate = 4; Sometimes accurate = 3; Occasionally accurate = 2; Never accurate = 1 |
| $a_{13}$ | Time required by the nearest disaster relief agency | Within 15 min = 4; 15–30 min = 3; 30–60 min = 2; Over 1 h = 1 |
| $a_{14}$ | Receipt of various forms of disaster prevention publicity and education | Yes = 1; No = 0 |
| $a_{15}$ | Regional rainfall | Over 250 mm/1 d = 5; 100 mm–250 mm/1 d = 4; 50 mm–100 mm/1 d = 3; 10 mm–50 mm/1 d = 2; Under 10 mm/1 d = 1; |

After the factors were assigned and all values were available, the data were processed according to the formulas in the TOPSIS method. The resilience levels of each community were calculated and ranked by dimensionless processing according to Equations (14) and (15), as shown in Table 8.

**Table 8.** Closeness degree and ranking.

| Community | $d_i+$ | $d_i-$ | $S_i$ | Ranking |
|---|---|---|---|---|
| Zhangjiawan community | 0.38 | 1.09 | 0.74 | 1 |
| Chuanchai community | 1.02 | 0.46 | 0.31 | 6 |
| Kangning community | 0.71 | 0.94 | 0.57 | 3 |
| Caiyuan community | 0.93 | 0.82 | 0.47 | 4 |
| Guanliu community | 0.92 | 0.59 | 0.39 | 5 |
| Pearl Garden | 0.58 | 0.99 | 0.63 | 2 |

The rankings can be summarized as the following: where the closeness degree $S_i$ is within the interval of 0.8–1, then the community resilience level is excellent and community residents are better off than other communities under the same circumstances; where the closeness degree $S_i$ is within the range of 0.6–0.8, then the community resilience level is good; where the closeness degree $S_i$ is within the range of 0.4–0.6, then the community resilience level is medium; where the closeness degree $S_i$ is within the range of 0.2–0.4, then the community resilience level is average; and where the closeness degree $S_i$ is below 0.2, then the community resilience level is poor and equally, residents of such communities are the most severely affected.

Through the empirical results (Figure 5), it can be seen that the resilience level of each community is sorted in the order of Zhangjiawan > Pearl Garden > Kangning > Caiyuan > Guanliu > Chuanchai. Zhangjiawan community resilience scored the highest, at 0.74, indicating a strong ability to resist flood risks. Due to the frequent occurrence of flood disasters in Wuhan, the municipal government has increased investment in urban flood control facilities and corresponding public infrastructure construction. These measures have effectively improved the community's ability to resist flood disasters. These results

are consistent with the actual situation of Wuhan's disaster prevention and mitigation investment and rescue capacity.

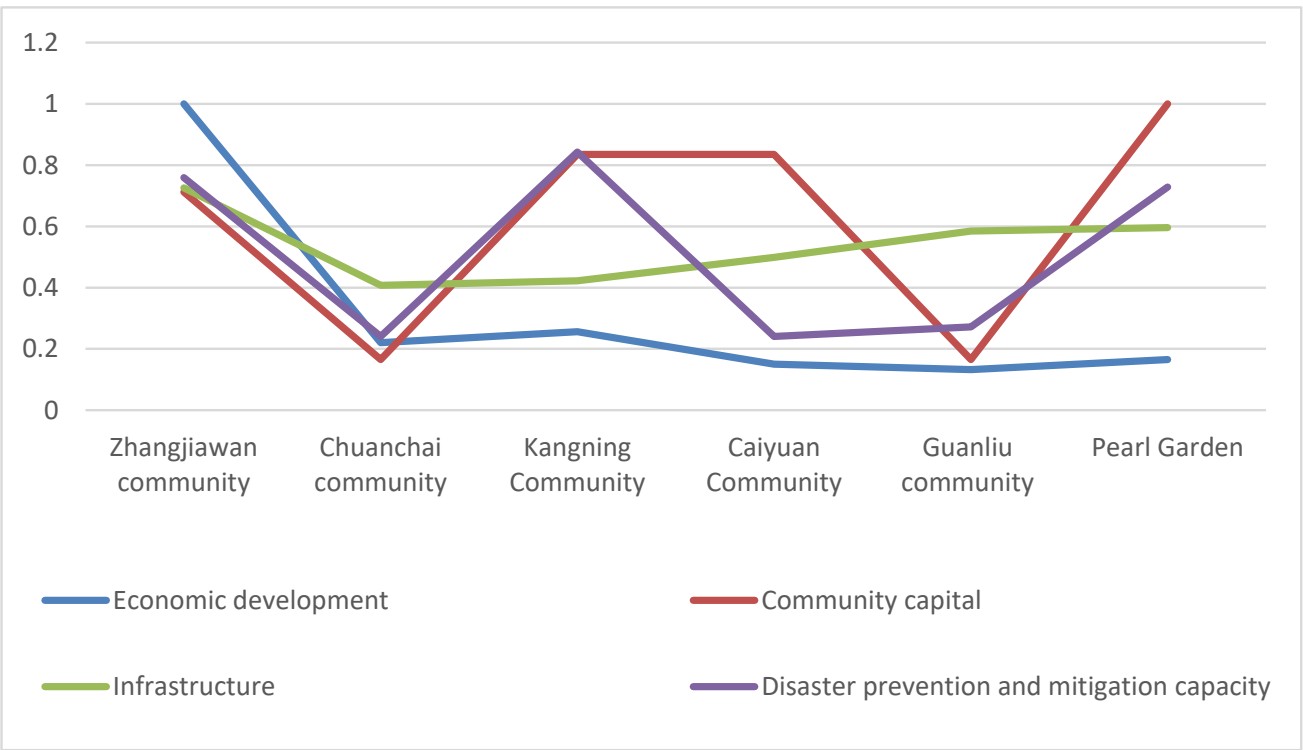

**Figure 5.** Comparison of six communities in the same dimension.

The Chuanchai community had the lowest resilience score of 0.31, indicating a low level of resilience to withstand floods. The Chuanchai community has an aging residential building stock, and the level of economic development, disaster prevention and mitigation capacity, infrastructure level and community capital are at a low level, not enough to cope with sudden disasters. In particular, the poor drainage capacity and low disaster relief capacity suggest that, in the event of severe flooding, they could bring about unimaginable consequences.

According to Figure 5, the resilience of the four dimensions of economic development, community capital, infrastructure and disaster prevention and mitigation capacity of the six communities can be intuitively analyzed. In the dimension of economic development, the resilience score of the Zhangjiawan community is the highest, followed by the Kangning community and the Chuanchai community. It can be judged that the economic development of the Zhangjiawan community is relatively rapid, and the three indicators of income level, medical insurance coverage and disaster prevention investment are at high levels. The economic development of the Kangning and Chuanchai communities are relatively slow, so they are vulnerable to the impacts of flooding. In the dimension of community capital, Pearl Garden has the highest resilience score and can effectively cope with sudden floods. The Kangning community, Caiyuan community and Zhangjiawan community follow, but the gap is not large.

In terms of infrastructure, the six communities are not too different, with the infrastructure of the Zhangjiawan community being the highest, followed by the Guanliu community and the Pearl Garden. Each community should improve its levels of infrastructure construction and ability to resist floods. In the dimension of disaster prevention and mitigation capacity, the Kangning community, Zhangjiawan community and Pearl Garden have higher resilience scores; that is to say, these three communities are superior to other communities in terms of flood discharge capacity, disaster relief capacity, disaster prevention publicity and education.

## 5. Discussion and Suggestions

### 5.1. Discussion

When flooding occurs, the community is the basic line of urban disaster prevention; so, we need to understand the factors that affect the resilience of the community to floods and identify key factors, non-key factors and current levels of resilience. In this study, the DEMATEL-ISM method was used to screen and sort 15 factors affecting the community's resilience to floods, and TOPSIS was used to calculate the resilience level of the selected community and sort the indicators to verify whether they were consistent with the results of the DEMATEL-ISM.

(1)    Methods

The DEMATEL can confirm the interdependence between the various factors and help to draw a map reflecting the relative relationship between the various factors, which can be used to investigate and solve complex and intertwined problems. This method not only transforms the interdependence relation into a causality group by matrix, but also finds the key factors of complex structural systems by producing an influence diagram. This method can divide the influencing factors into causal factors and result factors and identify the key influencing factors [50]. Among the 15 factors screened, the highest degree of centrality is disaster rescue capability ($a_{13}$), followed by disaster prevention publicity and education ($a_{14}$) and disaster prevention investment ($a_7$). The higher the degree of centrality, the more critical the factors are; so, these factors are also the key influencing factors. Although the DEMATEL method can obtain the importance of factors, other criteria are not used in the calculation and the relative weights of experts are not considered. Therefore, we chose to use a combination of the DEMATEL and ISM methods to set the threshold and to reduce the impact of individual experts on the overall results [51].

In this study, three methods were used to calculate the data continuously to ensure the reliability of the data operation. The use of these three methods is relatively rare in the literature where more commonly, either only the DEMATEL-ISM [52] or ISM-TOPSIS [53] or ANP-TOPSIS [54] methods have been used. The accuracy of the DEMATEL-ISM method was verified by using TOPSIS, and the resilience level of the community was evaluated and ranked [53]. The highest and lowest resilience communities in the selected communities were obtained, and the scores of each dimension were analyzed.

(2)    Community selection

Flooding is the most common natural hazard in China and in recent years, severe flooding has occurred frequently. As the smallest unit of the city, the community is an important field to resisting emergencies in a risky society and is key to preventing and responding to all kinds of emergencies. Communities also play an important role in the post-recovery phase. They play an important basic role in the whole cycle of the emergency management system. Community resilience has received extensive attention from international scholars in the development and improvement of risk management systems [55] and therefore, this research takes the community as the basic unit of analysis.

This research selected six communities, all located in Hubei Province, but belonging to different urban areas. Among them, four communities, the Zhangjiawan community in Wuhan City, Chuanchai community in Yichang City, Kangning community in Huangshi City and Caiyuan community in Suizhou City, have experienced flood disasters, while the remaining two communities, the Guanliu community in Ezhou City and Pearl Garden in Enshi City, have not. The types of communities are also different. The selected community types belong to two types: unit housing reform and commercial housing [43]. Unit housing reform refers to public housing built with the support of national policy. It is approved by the local government's housing reform or approved by the higher authorities of the selling unit and reported to the local government's housing reform for record [56,57]. Commercial housing in China emerged in the 1980s and refers to the construction of housing by real estate development and management companies (including foreign companies), approved by government agencies. This allows companies to rent land using rights periods of

40 years, 50 years, 70 years and involves the construction of housing and the sale of rental housing, including residential and commercial housing and other buildings.

(3)     Limitations

First, it is evident in the literature that the DEMATEL-ISM model is often used to study the factors or key factors of a particular aspect. However, it should be recognized that the initial data obtained when using this method is obtained from the views of experts. Hence, the presence of some subjectivity in the initial data is difficult to eliminate. Secondly, when using the TOPSIS method, only four of the dimensions for the six communities could be compared and ranked, and the specific ranking for the more detailed factors could not be derived. If further validation is needed, further research will be necessary using alternative methods. Finally the model and method proposed in this article are only validated in these six communities. Further research will be needed to test whether these approaches are more widely suitable in other cities or communities [58].

*5.2. Suggestions*

In summary, based on the analysis and comparison of 15 indicators and six communities using the results of the DEMATEL, ISM and TOPSIS method calculations, the following recommendations were made:

(1)     Raise the community's attention to community flood resilience. The community is the most basic urban unit and the first basic line of defense in terms of disaster response, with obvious vulnerability and sensitivity, and is most affected by various types of disasters. Strengthening community resilience to floods, improving people's disaster preparedness and resilience, and enhancing community self-help capabilities are the keys to reducing disaster losses.

Therefore, it is necessary to strengthen each society's attention to its community's flood resilience and improve each community's attention to the community's flood resilience, not only to improve social attention and increase the community's sense of existence for the city, but also to strengthen the government's sense of inspection and attention to the community and strengthen the community's governance capacity and flood resilience;

(2)     Increase government funding for community investment in flood resilience to improve the resilience of communities to floods. According to the TOPSIS method for community analysis, it can be seen that the Zhangjiawan community has a high level for its disaster prevention funding investment index and the highest community flood resilience. This indicates that the primary need is disaster prevention financial investment to improve community resilience to flooding. Disaster prevention funding directly affects community flood resilience, not only by influencing the construction of appropriate public infrastructure, but also by influencing the purchase and placement of community infrastructure for flood resilience and by having an impact on residents' acceptance of disaster prevention publicity and education, as well as other aspects [28]. Therefore, the government's investment in community funding for disaster preparedness affects not only the resilience of the economic development dimension of the community, but also the improvement of the infrastructure dimension of resilience and the disaster prevention and mitigation capacity dimension of resilience;

(3)     Strengthen the construction of the community's disaster prevention and mitigation capacity. The government's investment in disaster prevention affects the resilience of communities to floods and also affects the construction of community disaster prevention and mitigation capacity. The construction of community disaster prevention and mitigation capacity also affects the resilience of communities to floods. In the analysis of 15 indicators in six communities, the disaster prevention and mitigation capacity dimension of disaster rescue capacity and disaster prevention and public education are the key indicators that affect community resilience to floods. Therefore, community workers and people who maintain the normal order of the community must have certain disaster rescue capacity or have training about disaster rescue, and

community residents need to receive disaster prevention and publicity education regularly to improve their awareness of the community's needs to be educated regularly on disaster prevention and awareness.

## 6. Conclusions

This research has constructed a comprehensive model of community flood resilience. This has been developed following a detailed review of community flood resilience, based on an evaluation of the definitions of resilience and community resilience in the literature. The existing tools and frameworks have also been reviewed to assess their applicability, and based on their attributes, DEMATEL-ISM method and TOPSIS method were employed. Furthermore, the research has investigated the selection of indicators in four thematic areas, namely community capital, economic development, infrastructure and disaster prevention and mitigation. 15 indicators were identified around these thematic areas and used to construct an impact system of community flood resilience. The results from the DEMATEL-ISM method were then analyzed and subsequently validated to derive the underlying factors and the key factors found to affect community resilience to flooding. Then using the TOPSIS method, six communities in China were evaluated and ranked comprehensively, revealing that overall levels of community resilience were still at a relatively low level. The four dimensions were analyzed through the use of DEAMTEL-ISM method and TOPSIS method, while the indicators were classified in importance and the results were verified with each other. These findings are conducive to strengthening the sustainability of urban communities and provide useful scientific guidance for improving community resilience and in supporting strategic decision making in response to flooding.

Hence, this research has developed a comprehensive model of community flood resilience using a combination of both qualitative and quantitative methods. The impact of community flood resilience has been analyzed using the DEMATEL method, enabling the causal relationships between indicators, as well as the importance of indicators to be determined. Using the ISM method to classify the indicators, we further determined an improved understanding of the relationship between indicators. The TOPSIS method was then used to integrate a variety of risk indicators into a single overall score, which is convenient for the ranking of flood resilience.

Using this approach, and applying this to six communities in China, has enabled the following conclusions to be drawn:

(1) The DEMATEL-ISM method was used to analyze and study the relationship between each index. These results show that the indicators are divided into cause and result indicators, among which there are seven cause indicators and eight result indicators. According to the centrality of the index ranking, disaster relief ability is the most important, followed by disaster prevention education publicity and disaster prevention funds;

(2) The ISM method was used to divide the indicators into three levels and the directed graph was drawn to highlight the relationship between the layers. A sense of belonging, the age of the house and the place of public refuge were found to be the surface influencing factors. The investment in disaster prevention funds, disaster rescue ability and flood control and drainage ability were found to be the fundamental factors. The transition layer was found to include community relations, organizational activities, medical insurance coverage, medical service ability, income level, monitoring and early warning ability, community mutual assistance, convenient transportation ability, disaster prevention propaganda and education;

(3) An empirical study combined with the TOPSIS method were used to comprehensively evaluate and rank six communities: the Zhangjiawan community, the Chuanchai community, the Kangning community, the Caiyuan community, the Guanliu Community and the Peace Garden. The study found that the Zhangjiawan community, located in Wuhan City, had the highest levels of flood resilience and some of the management measures adopted in this community were highlighted as good practices. However, the overall level of community resilience was still found to be at a relatively low level.

Further research is recommended to develop and improve the key indicators and to identify measures to improve overall community resilience.

**Author Contributions:** Data curation, W.X. and Q.Y.; Writing, original draft, Y.X. and Q.Y.; Writing, review & editing, D.P. All authors have read and agreed to the published version of the manuscript.

**Funding:** The research was funded by the National Natural Science Foundation of Hubei Province [grant number, 2022CFC067].

**Data Availability Statement:** The meteorological data in the six communities were provided by the China Statistical Yearbook, the National Bureau of Statistics and the Chinese government's website. The authors are grateful for this support.

**Conflicts of Interest:** The authors declare no conflict of interest.

## Appendix A

**Table A1. Composite impact matrix C.**

|  | $a_1$ | $a_2$ | $a_3$ | $a_4$ | $a_5$ | $a_6$ | $a_7$ | $a_8$ | $a_9$ | $a_{10}$ | $a_{11}$ | $a_{12}$ | $a_{13}$ | $a_{14}$ | $a_{15}$ |
|---|---|---|---|---|---|---|---|---|---|---|---|---|---|---|---|
| $a_1$ | 0.296 | 0.403 | 0.354 | 0.411 | 0.312 | 0.359 | 0.375 | 0.360 | 0.298 | 0.304 | 0.369 | 0.435 | 0.427 | 0.443 | 0.412 |
| $a_2$ | 0.344 | 0.284 | 0.319 | 0.347 | 0.280 | 0.298 | 0.337 | 0.325 | 0.291 | 0.273 | 0.334 | 0.394 | 0.358 | 0.375 | 0.344 |
| $a_3$ | 0.360 | 0.359 | 0.280 | 0.365 | 0.271 | 0.341 | 0.356 | 0.344 | 0.281 | 0.315 | 0.353 | 0.390 | 0.434 | 0.422 | 0.368 |
| $a_4$ | 0.315 | 0.363 | 0.317 | 0.289 | 0.282 | 0.322 | 0.337 | 0.299 | 0.291 | 0.298 | 0.306 | 0.340 | 0.382 | 0.373 | 0.368 |
| $a_5$ | 0.364 | 0.387 | 0.391 | 0.424 | 0.266 | 0.425 | 0.413 | 0.375 | 0.335 | 0.317 | 0.383 | 0.422 | 0.417 | 0.455 | 0.399 |
| $a_6$ | 0.323 | 0.321 | 0.326 | 0.354 | 0.289 | 0.277 | 0.345 | 0.360 | 0.299 | 0.309 | 0.343 | 0.374 | 0.395 | 0.358 | 0.380 |
| $a_7$ | 0.359 | 0.416 | 0.364 | 0.450 | 0.344 | 0.396 | 0.359 | 0.423 | 0.359 | 0.368 | 0.434 | 0.477 | 0.503 | 0.487 | 0.512 |
| $a_8$ | 0.333 | 0.381 | 0.334 | 0.362 | 0.295 | 0.339 | 0.328 | 0.285 | 0.306 | 0.314 | 0.350 | 0.385 | 0.403 | 0.392 | 0.387 |
| $a_9$ | 0.368 | 0.372 | 0.371 | 0.405 | 0.303 | 0.379 | 0.421 | 0.407 | 0.287 | 0.380 | 0.419 | 0.431 | 0.484 | 0.442 | 0.466 |
| $a_{10}$ | 0.313 | 0.335 | 0.290 | 0.316 | 0.280 | 0.321 | 0.308 | 0.321 | 0.264 | 0.241 | 0.357 | 0.364 | 0.407 | 0.371 | 0.368 |
| $a_{11}$ | 0.323 | 0.346 | 0.326 | 0.354 | 0.262 | 0.331 | 0.347 | 0.307 | 0.300 | 0.308 | 0.287 | 0.376 | 0.423 | 0.385 | 0.383 |
| $a_{12}$ | 0.330 | 0.356 | 0.333 | 0.362 | 0.270 | 0.338 | 0.354 | 0.340 | 0.28 | 0.315 | 0.349 | 0.33 | 0.432 | 0.394 | 0.417 |
| $a_{13}$ | 0.333 | 0.387 | 0.364 | 0.396 | 0.321 | 0.369 | 0.413 | 0.372 | 0.335 | 0.343 | 0.382 | 0.421 | 0.389 | 0.457 | 0.480 |
| $a_{14}$ | 0.350 | 0.377 | 0.354 | 0.385 | 0.286 | 0.358 | 0.376 | 0.361 | 0.324 | 0.308 | 0.37 | 0.461 | 0.458 | 0.363 | 0.443 |
| $a_{15}$ | 0.289 | 0.338 | 0.318 | 0.345 | 0.281 | 0.298 | 0.339 | 0.299 | 0.293 | 0.300 | 0.308 | 0.368 | 0.412 | 0.402 | 0.318 |

**Table A2. Decomposition of influencing factors.**

|  | R | A | S = R ∩ A |
|---|---|---|---|
| $a_1$ | 1,2,3,4,12,14,15 | 1,3,4,7,12,13,14,15 | 1,3,4,12,14,15 |
| $a_2$ | 2,3,4,9,12,14,15 | 1,2,3,4,7,8,9,12,13,14,15 | 2,3,4,9,12,14,15 |
| $a_3$ | 1,2,3,4,5,12,14 | 1,2,3,4,7,12,13,14,15 | 1,2,3,4,12,14 |
| $a_4$ | 1,2,3,4,9,14 | 1,2,3,4,7,12,13,14,15 | 1,2,3,4,14 |
| $a_5$ | 5,6,8,12 | 3,5,6,10,12,13,14,15 | 12,5,6 |
| $a_6$ | 5,6,9 | 5,6,7,12,13,14,15 | 5,6 |
| $a_7$ | 1,2,3,4,6,7,8,11,12,13,14,15 | 7,12,13,14,15 | 7,12,13,14,15 |
| $a_8$ | 2,8,12,13,14,15 | 5,7,8,12,13,14,15 | 8,12,13,14,15 |
| $a_9$ | 2,9 | 2,4,6,9,12,13,14,15 | 9,2 |
| $a_{10}$ | 5,10,12,13,14,15 | 10,12,13,14,15 | 10,12,13,14,15 |
| $a_{11}$ | 11,12,13,14,15 | 7,11,12,13,14,15 | 11,12,13,14,15 |
| $a_{12}$ | 1,2,3,4,5,6,7,8,9,10,11,12,13,14,15 | 1,2,3,5,7,8,10,11,12,13,14,15 | 1,2,3,5,7,8,10,11,12,13,14,15 |
| $a_{13}$ | 1,2,3,4,5,6,7,8,9,10,11,12,13,14,15 | 7,8,10,11,12,13,14,15 | 7,8,10,11,12,13,14,15 |
| $a_{14}$ | 1,2,3,4,5,6,7,8,9,10,11,12,13,14,15 | 1,2,3,4,7,8,10,11,12,13,14,15 | 1,2,3,4,7,8,10,11,12,13,14,15 |
| $a_{15}$ | 1,2,3,4,5,6,7,8,9,10,11,12,13,14,15 | 1,2,7,8,10,11,12,13,14,15 | 1,2,7,8,10,11,12,13,14,15 |

**Table A3.** Community description.

| Geographic Position | Wuhan City | Yichang City | Huangshi City | Suizhou City | Ezhou City | Enshi City |
|---|---|---|---|---|---|---|
| Community name | Zhangjiawan community | Chuanchai community | Kangning community | Caiyuan community | Guanliu community | Pearl Garden |
| Year of construction | 1994 | 1985 | 1975 | 2000 | 1988 | 1996 |
| type | Commercial building | Unit room change room | Unit room change room | Commercial building | Commercial building | Commercial building |
| Have experienced floods | Yes | Yes | Yes | Yes | No | No |
| Description of community characteristics | High vegetation coverage, good community environment, general management | Housing age, inadequate infrastructure, mainly for the resettlement of plant staff accommodation | Old and poor infrastructure, aging public facilities, mainly workers and their families | Dense population, more harmonious community relations | Large built area, long house age | Better community relations, better infrastructure |

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
