# Peer review of "An Evaluation of Factors Influencing the Resilience of Flood-Affected Communities in China"

_hydrology, doi:10.3390/hydrology10020035_

Round 1

Reviewer 1 Report

First of all, thank you for the opportunity to do a review. The following are my comments regarding the overall reception of the article, its layout and its merit.

The introduction is very well and interestingly written. It contains a consistant review of the literature in the subject of flood research and introduces the necessary state-of-art. 

Dematel system written in an understandable manner. ISM and TOPSIS system description structure also correct and understandable. There are a few errors involving omitted spaces in the text.

Do the proposed methods have any weaknesses / limitations in application? This is certainly an interesting issue for the reader, however, it is not a prerequisite for accepting the text for publication. 

Lines 214-366 I think that there is a lack of appropriate formatting, the reception of the text quite chaotic. Chapter 3 written substantively correct and understandable.

Fig. 5. color coding should be changed, as the figure is poorly legible. It would be possible to combine Fig. 4 and 5 in parts a and b on one line.

The literature list-although quite dogmatic-is nevertheless limited to, for the most part, local sources. The literature list should be standardized.

Reviewer 3 Report

1. The authors should use a uniform style of reference in the paragraph. Such as Line 85 Zhang R-Y, Line 91 Tan S-B, Line 101 Chen et al., Line 104 Li Ya et al., and Line 112 Wu Huaqing et al.

2. As in the introduction part, the authors list the literature assessment resilience, but did not point out the disadvantage of these works, and did not analyze the relationship between the existing works and this manuscript, thus the meaning of this manuscript is not sufficient enough.

3. The DEMATEL-ISM method has been used for a long time, the literature can be found in 2008, such as Zhou D, Zhang L. Establishing hierarchy structure in complex systems based on the integration of DEMATEL and ISM[J]. Journal of Management Sciences in China, 2008, 11(2):20-26. And the authors should introduce their own contributions to this manuscript, and this may increase the readability of this work.

4. Please check the correctness of the formulas carefully, such as Eq-3, Eq-4, and Eq-5. And all the parameters used in the formulas should be explained.

Round 2

Reviewer 3 Report

It can be accepted.